# Identifying the Limits of Cross-Domain Knowledge Transfer for Pretrained Models

## Abstract

There is growing evidence that pretrained language models improve task-specific fine-tuning even where the task examples are radically different from those seen in training. What is the nature of this surprising cross-domain transfer? We offer a partial answer via a systematic exploration of how much transfer occurs when models are denied any information about word identity via random scrambling. In four classification tasks and two sequence labeling tasks, we evaluate LSTMs using GloVe embeddings, BERT, and baseline models. Among these models, we find that only BERT shows high rates of transfer into our scrambled domains, and for classification but not sequence labeling tasks. Our analyses seek to explain why transfer succeeds for some tasks but not others, to isolate the separate contributions of pretraining versus fine-tuning, to show that the fine-tuning process is not merely learning to unscramble the scrambled inputs, and to quantify the role of word frequency. These findings help explain where and why cross-domain transfer occurs, which can guide future studies and practical fine-tuning efforts.

## 1 Introduction

Fine-tuning pretrained language models has proven to be highly effective across a wide range of NLP tasks; the leaderboards for standard benchmarks are currently dominated by models that adopt this general strategy (Rajpurkar et al., 2016; 2018; Wang et al., 2018; Yang et al., 2018; Wang et al., 2019). Recent work has extended these findings in even more surprising ways: Artetxe et al. (2020), Karthikeyan et al. (2019), and Tran (2020) find evidence of transfer between natural languages, and Papadimitriou & Jurafsky (2020) show that pretraining language models on non-linguistic data such as music and computer code can improve test performance on natural language.

Why does pretraining help even across what appear to be fundamentally different domains, and what are the limits of such cross-domain transfer? In this work, we seek to inform these questions via a systematic exploration of how much cross-domain transfer we see when the model is denied any information about word identity. In this setting, we can vary the pretraining and fine-tuning examples dramatically while holding other aspects of the task constant. This allows us to quantify the extent of transfer, and it can yield insights into the wide-ranging transfer results cited above.

Figure 1 gives an overview of our core experimental paradigm: starting with two identical copies of a single pretrained model for English, we fine-tune one on English examples and the other on scrambled English sentences, using a scrambling function $F$ (section 3), and then we evaluate the resulting models. We apply this paradigm to four classification tasks and two sequence modeling tasks, and we evaluate bag-of-words baselines, LSTMs with GloVe initialization and rich attention mechanisms, and BERT. Our central finding is that only BERT is able to achieve robust cross-domain transfer, and for classification tasks but not sequence labeling ones.

To try to understand why such transfer is successful for some tasks but not others, we pursue a number of hypotheses. First, we consider whether using a scrambling function $F$ that matches word frequencies is required for transfer, and we find that such matching plays a small role, but not enough to account for the observed performance (section 7.1). Second, we assess whether frequency matching might actually be inserting semantic consistency into the scrambling process by, for example, systematically creating substitution pairs like *good/great* and *professor/teacher* (section 7.2). However, we find no evidence of such semantic consistency. Third, we try to isolate the contribution of pretraining versus fine-tuning by fine-tuning randomly initialized models of different

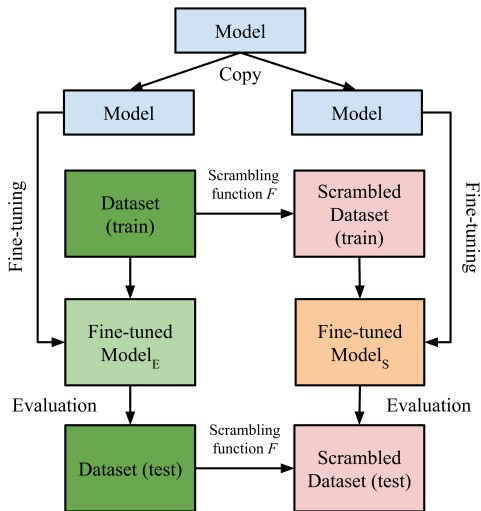

Figure 1: An overview of our experiment paradigm. Starting with a model (e.g., pretrained BERT, GloVe-initialized LSTM, etc.), we copy it and fine-tune it on the regular and scrambled train set using a scrambling function $F$. The model is then evaluated on regular and scrambled test sets. Our paper explores different options for $F$ and a number of variants of our models to try to quantity the amount of transfer and identify its sources.

sizes (section 7.3) and by freezing the BERT parameters, such that only task-specific parameters are updated (section 7.4). These variations lead to a substantial drop in transfer, suggesting that fine-tuning is vital, although our LSTM results show that the BERT pretrained starting point is also an essential component. Fourth, we ask whether the fine-tuning process is primarily learning to reassociate scrambled words with their sources, and we find that it is not (section 7.5). While these findings do not fully account for the transfer we observe, they offer a partial explanation which should help guide future studies of this issue and which can help with practical fine-tuning work.

## 2    RELATED WORK

### 2.1    EVIDENCE FOR TRANSFER

Transferability across domains is often used to benchmark large pretrained models such as BERT (Devlin et al., 2019b), RoBERTa (Liu et al., 2019b), ELECTRA (Clark et al., 2019), and XL-Net (Yang et al., 2019). To assess transferability, pretrained models are fine-tuned for diverse downstream tasks (Wang et al., 2018; 2019). Recently, pretrained Transformer-based models (Vaswani et al., 2017) have even surpassed estimates of human performance on GLUE (Wang et al., 2018) and SuperGLUE (Wang et al., 2019). While the benefits of pretraining are reduced when there is a large train set (Hernandez et al., 2021), there is little doubt that this pretraining process helps in many scenarios.

### 2.2    STUDIES OF WHY TRANSFER HAPPENS

There are diverse efforts underway to more deeply understand why transfer occurs. Probing tests often involve fitting supervised models on internal representations in an effort to determine what they encode. Such work suggests that BERT representations encode non-trivial information about morphosyntax and semantics (Tenney et al., 2019; Liu et al., 2019a; Hewitt & Manning, 2019; Manning et al., 2020) and perhaps weakly encode world knowledge such as relations between entities (Da & Kasai, 2019; Petroni et al., 2019), but that they contain relatively little about pragmatics or role-based event knowledge (Ettinger, 2020). Newer feature attribution methods (Zeiler & Fergus, 2014; Springenberg et al., 2015; Shrikumar et al., 2017; Binder et al., 2016; Sundararajan et al., 2017) and intervention methods (McCoy et al., 2019; Vig et al., 2020; Geiger et al., 2020) are corroborating these findings while also yielding a picture of the internal causal dynamics of these models.

| Scrambling Method | Sentence |
|---|---|
| Original English (No Scrambling) | "the worst titles in recent cinematic **history**" |
| Similar Frequency | "a engaging semi is everyone dull **dark**" |
| Random | "kitsch theatrically tranquil andys loaf shorty **lauper**" |

Table 1: An example from the SST-3 dataset and its two scrambled variants.

Another set of strategies for understanding transfer involves modifying network inputs or internal representations and studying the effects of such changes on task performance. For instance, Tamkin et al. (2020) show that BERT's performance on downstream GLUE tasks suffers only marginally even if some layers are reinitialized before fine-tuning, and Gauthier & Levy (2019), Zanzotto et al. (2020), Pham et al. (2020), and Sinha et al. (2021) show that BERT-like models are largely insensitive to word order changes.

### 2.3 Extreme Cross-Domain Transfer

Cross-domain transfer is not limited to monolingual cases (Karthikeyan et al., 2019). With modifications to its tokenizer, English-pretrained BERT improves performance on downstream multilingual NLU tasks (Artetxe et al., 2020; Tran, 2020). Papadimitriou & Jurafsky (2020) show that pretraining language models on structured non-linguistic data (e.g., MIDI music or Java code) improves test performance on natural language. Our work complements and advances these efforts along two dimensions. First, we challenge models with extremely ambitious cross-domain settings and find that BERT shows a high degree of transfer, and we conduct a large set of follow-up experiments to help identify the sources and limitations of such transfer.

## 3 Experimental Paradigm

We now describe the evaluation paradigm summarized in figure 1 (section 3.1), with special attention to the scrambling functions $F$ that we consider (sections 3.2–3.3).

### 3.1 Evaluation Pipeline

Figure 1 shows our main evaluation paradigm for testing the transferability of a model without word identity information. On the left side, we show the classic fine-tuning pipeline (i.e., we fine-tune on the original English training set and evaluate on the original English test set). On the right side, we show our new evaluation pipeline: starting from a single model, we (1) fine-tune it with a corrupted training split where regular English word identities are removed and then (2) evaluate the model on a version of the evaluation set that is corrupted in the same manner. The paradigm applies equally to models without any pretraining and with varying degrees of pretraining for their model parameters.

### 3.2 Scrambling with Similar Frequency

To remove word identities, we scrambled each sentence in each dataset by substituting each word $w$ with a new word $w'$ in the vocabulary of the dataset. For Scrambling with Similar Frequency, we use the following rules:

1. $w$ and $w'$ must have the same sub-token length according to the BERT tokenizer; and
2. $w$ and $w'$ must have similar frequency.

The first rule is motivated by the concern that sub-token length may correlate with word frequency, given that rarer and longer words may be tokenized into longer sub-tokens. The second rule is the core of the procedure. The guiding idea is that word frequency is often reflected in learned embeddings (Gong et al., 2018), so this scrambling procedure might preserve useful information and thus help to identify the source of transfer. Table 5 shows an example, and Appendix C provides details about the matching algorithm and additional examples of scrambled sentences.

| Dataset | Standard Models (Train and Test on English) | | | | Scrambled Models (Train and Test on Scrambled English) | | | |
|---------|------|------|-----|-------|------|------|------|------|
| | BERT | LSTM | BoW | Dummy | BERT-Scrambled Similar Frequency | Random | LSTM-Scrambled Similar Frequency | Random |
| **SST-3** | .71 (.02) | .62 (.01) | .59 (.00) | .33 (.02) | .65 (.01) | .64 (.02) | .57 (.02) | .56 (.02) |
| **SNLI** | .91 (.02) | .78 (.02) | 66 (.02) | .33 (.01) | .84 (.01) | .82 (.02) | .72 (.00) | .71 (.01) |
| **QNLI** | .91 (.02) | .68 (.02) | .62 (.01) | .50 (.01) | .82 (.01) | .79 (.02) | .62 (.01) | .61 (.01) |
| **MRPC** | .86 (.01) | .72 (.02) | .70 (.02) | .50 (.02) | .82 (.02) | .78 (.02) | .69 (.00) | .68 (.00) |
| **EN-EWT** | .97 (.01) | .85 (.02) | .65 (.01) | .09 (.01) | .86 (.01) | .81 (.02) | .80 (.01) | .72 (.01) |
| **CoNLL-2003** | .95 (.01) | .75 (.01) | .28 (.02) | .02 (.01) | .74 (.01) | .72 (.02) | .61 (.02) | .56 (.01) |

Table 2: Model performance results for models trained on original English and on scrambled English. Standard deviations are reported for all entries.

## 3.3 RANDOM SCRAMBLING

To better understand the role of frequency in domain transfer, we also consider a word scrambling method that does not seek to match word frequencies. For this, we simply shuffle the vocabulary and match each word with another random word in the vocabulary without replacement. We include the distributions of the difference in frequency for every matched word pair in Appendix C to make sure a word is paired with a new word with drastically different frequency in the dataset. We also tried to pair words by the reverse order of frequencies, which yielded similar results, so we report only random scrambling results here.

## 4 MODELS

In this section, we describe the models we evaluated within our paradigm. Appendix B provides additional details about how the models were designed.

**BERT**   For our BERT model (Devlin et al., 2019a), we import weights from the pretrained BERT-base model through the HuggingFace `transformers` library (Wolf et al., 2020). For sequence classification tasks, we append a classification head after the `[CLS]` token embedding in the last layer of the BERT model. If an input example contains a pair of sentences, we concatenate them using a `[SEP]` token in between. For sequence labeling tasks, we append a shared classification head to each token embedding in the last layer of the BERT model.

**LSTM**   We contextualize our results against a strong LSTM-based model (Hochreiter & Schmidhuber, 1997). We lower-case each input sentence and tokenize it by separating on spaces and punctuation. We then use 300-dimensional GloVe embeddings (Pennington et al., 2014)[1] as inputs to a single-layer recurrent neural network with LSTM cells, with a hidden size of 64. We use dot-product attention (Luong et al., 2015) to formulate a context vector for each sentence. Finally, we pass the context vector through a multilayer perceptron (MLP) layer to get the final prediction. For an input example with a pair of sentences, we concatenate two sentences together before feeding them into our LSTM encoder. For sequence labeling tasks, we directly feed the hidden state at each position to the MLP layer to get the final prediction.

**Bag-of-Words (`BoW`) Model**   We compare against a BoW classifier, which serves as a proxy of model performance when only given word co-occurrence information. For each sentence in a dataset, we first formulate a BoW vector that uses unigram representations of an input sentence. Then, we feed the BoW vector through a softmax classifier. For examples with a pair of sentences, we create two BoW vectors for each sentence, and concatenate them together before feeding them into the linear layer for predicting labels. For sequence labeling tasks, we use Conditional Random Fields models (CRFs; Lafferty et al., 2001) with character-level unigram BoW features.

**Dummy Model**   We include a random classifier that generates predictions randomly proportional to the class distribution of the training set. We use this model to further contextualize our results.

---

[1]We use the Common Crawl cased version: `http://nlp.stanford.edu/data/glove.840B.300d.zip`

| Dataset | Type | #Train | #Dev | #Test | #Class |
|---|---|---|---|---|---|
| SST-3 | Sequence Classification | 159k | 1,1k | 2.2k | 3 |
| SNLI | Sequence Classification | 550k | 10k | 10k | 3 |
| QNLI | Sequence Classification | 108k | 5.7k | 5.7k | 2 |
| MRPC | Sequence Classification | 3.7k | 408 | 1.7k | 2 |
| EN-EWT UPOS | Sequence Labeling | 14k | 2k | 3.5k | 18 |
| CoNLL-2003 NER | Sequence Labeling | 12.5k | 2k | 2.1k | 9 |

Table 3: Summary information for each task.

## 5 TASKS

We consider six sequence classification and sequence labeling tasks (Table 3).

**Sequence Classification**   We select four NLU datasets for sequence classification. We consider sentiment analysis (SST-3; Socher et al., 2013), where SST-3 is a variant of the Stanford Sentiment Treebank with positive/negative/neutral labels; we train on the phrase- and sentence-level sequences in the dataset and evaluate only on its sentence-level labels. Additionally, we include natural language inference (QNLI; Demszky et al., 2018 and SNLI; Bowman et al., 2015) and paraphrase (MRPC; Dolan & Brockett, 2005). QNLI is derived from a version of Stanford Question Answering Dataset. For sequence classification tasks, we use Macro-F1 scores for SST-3, and accuracy scores for other NLU tasks.

**Sequence Labeling**   In contrast to sequence classification, where the classifier only considers the [CLS] token of the last layer and predicts a single label for a sentence, sequence labeling requires the model to classify all tokens using their contextualized representations. We select two datasets covering distinct tasks: part-of-speech detection (POS) and named entity recognition (NER). We used Universal Dependencies English Web Treebank (EN-EWT)  (Silveira et al., 2014) for POS, and CoNLL-2003 (Tjong Kim Sang & De Meulder, 2003) for NER. For sequence labeling tasks, we used Micro-F1 (i.e., accuracy with full labels) for POS and F1 scores for NER.

## 6 RESULTS

In this section, we analyze the fine-tuning performance of BERT on scrambled datasets. Table 2 shows performance results. We focus for now on the results for Scrambling with Similar Frequency. Additionally, we also include baseline models trained with original sentences for comparison purposes. When training models on each task, we select models based on performance on the dev split during fine-tuning. We average performance results with multiple random seeds to get stabilized results. See Appendix B for additional details on our training and evaluation procedures.

### 6.1 SEQUENCE CLASSIFICATION

Comparing the second column (BERT model that is trained and tested on English) with the sixth column (BERT model that is trained and tested on Scrambled English with Similar Frequency Scrambling) in Table 2, we see that BERT maintains strong performance for all sequence classification tasks even when the datasets are scrambled. More importantly, we find that BERT fine-tuned with a scrambled dataset performs significantly better than the LSTM model (with GloVe embeddings) trained and evaluated on standard English data

For example, on the MRPC task, BERT evaluated with scrambled data experiences a less than 5% performance drop, and shows significantly better performance (a 13.9% improvement) than the best LSTM model. BERT evaluated with scrambled QNLI experiences the biggest drop (a 9.89% decrease). However, this still surpasses the best LSTM performance by a large margin (a 20.6% improvement).

Table 2 also presents performance results for other baseline models, which can be used to assess the intrinsic difficulty of each task. Our results suggest that BERT models fine-tuned with scrambled

| Dataset | LSTM-Baseline | LSTM-Scrambled Similar Frequency | |
| | | GloVe | No GloVe |
|---|---|---|---|
| **SST-3** | .62 (.01) | .57 (.02) | .58 (.01) |
| **SNLI** | .78 (.02) | .72 (.00) | .71 (.00) |
| **QNLI** | .68 (.02) | .62 (.01) | .61 (.01) |
| **MRPC** | .72 (.02) | .69 (.00) | .69 (.00) |
| **EN-EWT** | .85 (.02) | .80 (.01) | .79 (.01) |
| **CoNLL-2003** | .75 (.01) | .61 (.02) | .60 (.01) |

Table 4: Performance results for LSTM models trained on regular English and on English with Scrambling with Similar Frequency, with GloVe embeddings and with randomly initialized embeddings.

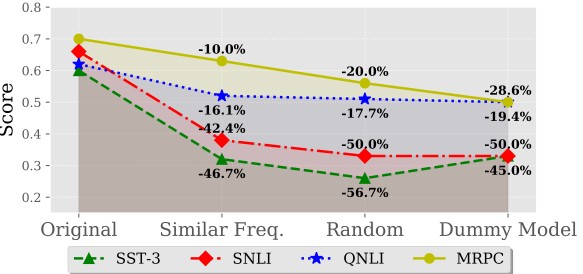

Figure 2: Zero-shot evaluation with the Bag-of-Word (BoW) model on scrambled datasets and the dummy model. Numbers are the differences between the current points and the first points in percentages.

tasks remain very strong across the board, and they remain stronger than best LSTM baseline models (those trained and tested on regular English) in all the classification tasks.

The overall performance of the LSTM models is worth further attention. The LSTMs are far less successful at our tasks than the BERT models. However, it seems noteworthy that scrambling does not lead to catastrophic failure for these models. Rather, they maintain approximately the same performance in the scrambled and unscrambled conditions. This might seem at first like evidence of some degree of transfer. However, as we discuss in section 7.3, the more likely explanation is that the LSTM is simply being retrained more or less from scratch in the two conditions.

## 6.2 Sequence Labeling

For a more complex setting, we fine-tuned BERT with sequence labeling tasks, and evaluated its transferability without word identities (i.e., using datasets that are scrambled in the same way as in our sequence classification tasks). The second set (bottom set) of Table 2 shows performance results for sequence labeling tasks where the goal of the BERT model is to classify every token correctly. As shown in Table 2, BERT experiences a significant drop when evaluated with a scrambled dataset for a sequence labeling task. For LSTMs trained with scrambled sequence labeling tasks, we also observe bigger drops compared with sequence classification tasks. For CoNLL-2003, LSTM with GloVe embeddings drops (a 18.7% decrease) from its baseline counterpart. Our results suggest that transfer learning without word identities is much harder for sequence labeling tasks. One intuition is that sequence labeling tasks are more likely to rely on word identities given the fact that classification (i.e., labeling) is at the token-level.

## 7 Analysis

### 7.1 Frequency Effects

Preserving word frequencies during scrambling may lead to higher performance when training and evaluating on scrambled datasets. To assess how much of the observed transfer relates to this factor, we can compare Scrambling with Similar Frequency (SSF) with Random Scrambling (RS), as described in section 3. As shown in Table 2, performance drops slightly if we use RS. For sequence classification tasks, RS experiences 1–5% drops in performance compared with SSF. For sequence labeling tasks, the difference is slightly larger: about 2–6%. This suggests that word frequency is indeed one of the factors that affects transferability, though the differences are relatively small, indicating that this is not the only contributing factor. This is consistent with similar findings due to Karthikeyan et al. 2019 for multilingual BERT.

### 7.2 Does Scrambling Preserve Meaning?

Another explanation is that our scrambling methods tend to swap words that are predictive of the same labels. For example, when we are substituting words with similar frequencies in SST-3,

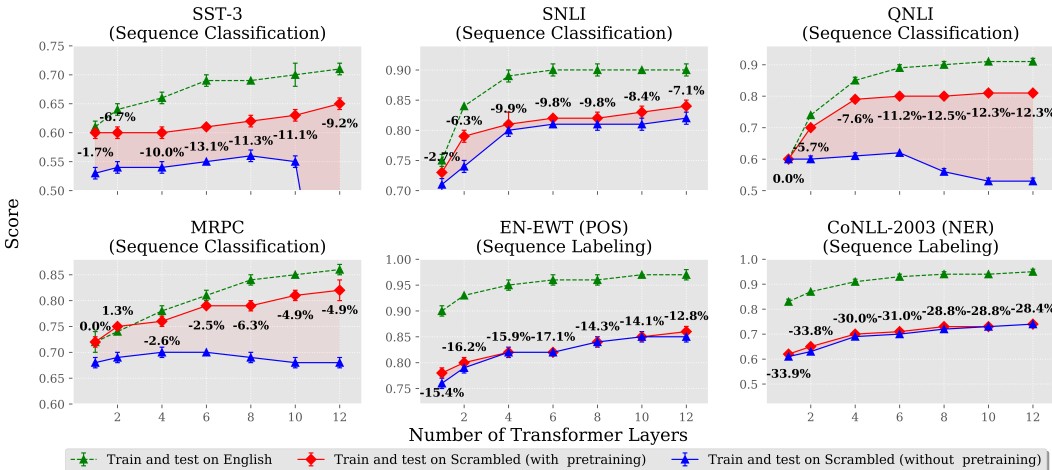

Figure 3: Performance results when fine-tuning end-to-end for different number of Transformer layers. Annotated numbers are the differences between the red lines and the green lines in percentages. Scoring for each task is defined in Section 5.

"good" may be swapped with "great" since they may have similar frequencies in a sentiment analysis dataset. To rule this out, we conducted zero-shot evaluation experiments with our BoW model on sequence classification tasks. The rationale here is that, to the extent that our swapping preserved the underlying connection between features and class labels, this should show up directly in the performance of the BoW model. For example, just swapping of "good" for "great" would hardly affect the final scores for each class. If there are a great many such invariances, then it would explain the apparent transfer.

Figure 2 shows the zero-shot evaluation results of our BoW model on all sequence classification datasets. Our results suggest that both scrambling methods result in significant performance drops, which suggests that word identities are indeed destroyed by our procedure, which again shines the spotlight on BERT as the only model in our experiments to find and take advantage of transferable information.

### 7.3    TRANSFER OR SIMPLE RETRAINING?

Our results on classification tasks show that English-pretrained BERT can achieve high performance when fine-tuned and evaluated on scrambled data. Is this high performance uniquely enabled by transfer from BERT's pretrained representations, or is BERT simply re-learning the token identities from its scrambled fine-tuning data?

To distinguish between these two hypotheses, we first examine whether randomly-initialized BERT models can also achieve high performance when fine-tuned and evaluated on scrambled data. We study models of varying capacity by modulating the number of BERT Transformer blocks. See Appendix B for details about the training procedure for these randomly-initialized models.

We compare these varying-depth randomly-initialized models against BERT models pretrained on English. To modulate the capacity of these pretrained models, we progressively discard the later Transformer layers (i.e., we make predictions from intermediate layers). Comparing these models is a step toward disentangling the performance gains of pretraining from the performance gains relating to model capacity.

Figure 3 summarizes these experiments. The red line represents our fine-tuning results, across different model sizes. The shaded area represents the performance gain from pretraining when training and testing on scrambled data. Pretraining yields consistent gains across models of differing depths, with deeper models seeing greater gains.

For sequence labeling tasks, the patterns are drastically different: the areas between the two lines are small. Since the random-initialized and pretrained models achieve similar performance when fine-

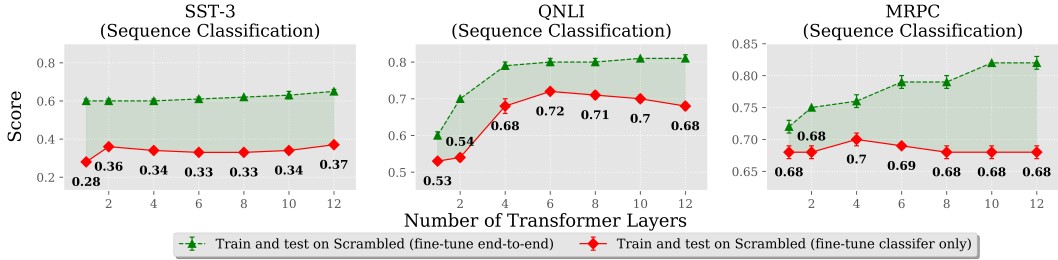

Figure 4: Performance results when fine-tuning only the classifier head by freezing all proceeding layers in BERT (red line) vs. fine-tuning end-to-end, which includes the classifier head and all proceeding layers in BERT (green line). Numbers are scores for the red lines. Scoring for each task is defined in Section 5.

tuned and tested on scrambled data, pretraining is not beneficial. This suggests that BERT hardly transfers knowledge when fine-tuned for sequence labeling with scrambled data.

Table 4 shows our results when training LSTMs without any pretrained embeddings. Unlike with BERT, GloVe initialization (a pretraining step) hardly impacts model performance across all tasks. Our leading hypothesis here is that the LSTMs may actually relearn all weights without taking advantage of pretraining. All of our LSTM models have parameter sizes around 1M, whereas the smallest BERT model (i.e., with a single Transformer layer) is around 3.2M parameters. Larger models may be able to rely more on pretraining.

Overall, these results show that we do see transfer of knowledge, at least for classification tasks, but that there is variation between tasks on how much transfer actually happens.

### 7.4 Assessing Transfer with Frozen BERT Parameters

We can further distinguish the contributions of pretraining versus fine-tuning by freezing the BERT parameters and seeing what effect this has on cross-domain transfer. Ethayarajh (2019) provides evidence that early layers are better than later ones for classifier fine-tuning, so we explore the effects of this freezing for all the layers in our BERT model.

As shown in Figure 4, performance scores drop significantly if we only fine-tune the classifier head and freeze the rest of the layers in BERT across three of our tasks. However, we find that performance scores change significantly depending on which layer we append the classifier head to. Consistent with Ethayarajh's findings, contextualized embeddings in lower layers tend to be more predictive. For example, if we freeze BERT weights and use the contextualized embeddings from the 2nd layer for SST-3, the model reaches peak performance compared with contextualized embeddings from other layers. More importantly, the trend of the green line follows the red line in Figure 4, especially for SST-3 and QNLI. The only exception is MRPC, where the red line plateaus but the green line keeps increasing. This could be an artifact of the size of the dataset, since MRPC only contains around 3.7K training examples. Our results suggest that pretrained weights in successive self-attention layers provide a good initial point for the fine-tuning process.

### 7.5 Probing for Word Identity Reassociations

We further investigate the learning dynamics of our fine-tuned models. Specifically, we study whether our fine-tuned models reassociate word identities with tokens for our sequence classification tasks. To do this, we measure the cosine similarities between words and their scrambled counterparts before and after the fine-tuning process.[2] To the extent that these similarities are increased after fine-tuning, we have evidence that fine-tuning has learned to ressociate words with their scrambled counterparts. However, we find essentially no evidence for such changes. As shown in figure 5, the correlation distributions before fine-tuning and after are extremely similar. This suggests that our fine-tuned models rarely reassociate word identities in the embedding layer.

---

[2]We only consider shared words in the model vocabulary and our scrambling maps, which includes 30% of words in the model vocabulary.

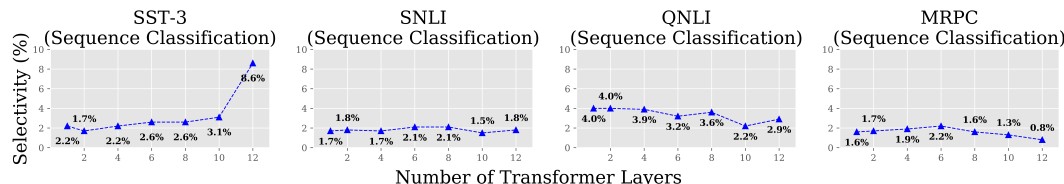

Figure 5: Correlations between cosine similarities of word embeddings before fine-tuning v.s. fine-tuning with scrambled datasets. Measurements of correlations are defined in Section 7.5.

Figure 6: Accuracy of word identity probes when applied to hidden states of each layer comparing to the control task introduced by Hewitt & Liang (2019). Measurements of accuracies are defined in Section 7.5.

To push this analysis a step further, we probe whether word identities are recovered through transformer layers by adapting the probing method with control task from Hewitt & Liang (2019). Formally, we use an MLP classifier to predict the word identity for $w$ using the contextualized hidden representations of its scrambled counterpart $w'$. For our control task, we ask the probe to predict random word identities. The difference in performance between these two conditions is know as *selectivity*, and it estimates the degree to which the word identities are recoverable, taking the power of the probe model into account. As shown in figure 5, our results suggest that relatively little information about the scrambling map is latent in these representations, across tasks and model layers.

# 8 CONCLUSION

In this paper, we propose an evaluation pipeline for pretrained models by testing their transferability without word identity information. Specifically, we take an English pretrained BERT off-the-shelf and fine-tune it with a scrambled English dataset. We conduct analyses across six tasks covering both classification and sequence labeling. By evaluating performance against multiple baselines, we aim to assess where BERT can transfer knowledge even without word identities. We find considerable transfer for BERT as compared to even powerful baselines, by only for classification tasks.

What is the source of successful cross-domain transfer with BERT? We find that word frequency contributes, but only to a limited extent: scrambling with matched word frequencies consistently outperforms scrambling with unmatched word frequencies, but transfer still occurs robustly even with random scrambling. We are also able to determine that both pretraining and fine-tuning are important and interacting factors in this transfer; freezing BERT weights during task-specific training leads to much less transfer, but too much task-specific training erodes the benefits of pretraining and in turn reduces the amount of transfer observed.

These analyses begin to piece together a full account of these surprising transfer results for BERT, but they do not fully explain our experimental results. Recent literature suggests at least two new promising avenues to explore. First, Sinha et al. (2021) seek to help characterize the rich distributional prior that models like BERT may be learning, which suggests that higher-order notions of frequency play a significant role in transfer. Second, the findings of Ethayarajh (2019) may be instructive: through successful layers, BERT seems to perform specific kinds of dimensionality reduction that help with low-dimensional classification tasks. Our results concerning layer-wise variation are consistent with this. And there may be other paths forward. The more we can learn about the extent of cross-domain transfer, the more effectively we can train and fine-tune these models on challenging tasks.

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

## APPENDIX FOR 'IDENTIFYING THE LIMITS OF CROSS-DOMAIN KNOWLEDGE TRANSFER FOR PRETRAINED MODELS'

## A  DATASETS

Table 3 in our main text shows statistics for the six datasets included in our experiments. We use the Dataset interface provided by the `Hugging Face` library Wolf et al. (2020) to foster reproducibility. For each scrambling test, we use the same splits as in the original datasets.

## B  MODEL AND TRAINING SETUP

**BERT Model**  Our BERT model has 12 heads and 12 layers, with hidden layer size 768. The model uses the WordPiece tokenizer, with a maximum sequence length of 128. We fine-tune our model

with a dropout probability of 0.1 for both attention weights and hidden states. We employ early stopping with a patience of 5. This ensures a fair comparison between different settings.

We use original BERT Adaam optimizer (Kingma & Ba, 2014) with the default cross-entropy loss as our loss function. Through our experiments, we discover the initial learning rate plays an important role for performance across all datasets. Thus, we optimize over a wide range of initial learning rates including $\{2e^{-5}, 4e^{-5}, 6e^{-5}, 8e^{-5}, 1e^{-4}, 4e^{-4}, 8e^{-4}\}$. For each initial learning rate, we repeat our experiments for 3 different random seeds. Table 2 shows the best averaged performance. To foster reproducibility, our training pipeline is adapted from the `Hugging Face` library Wolf et al. (2020). We use 6 × GeForce RTX 2080 Ti GPU each with 11GB memory. The training process takes about 1 hour to finish for the largest dataset and 15 minutes for the smallest dataset.

**`LSTM` Model** Similar to our BERT model, we use a maximum sequence length of 128. We employ a training batch of 1024, and early stopping with a patience of 5. This ensures a fair comparison between different settings. It is worth to noting that we find that BERT converges with scrambled datasets as quickly as (i.e., with same amount of steps) fine-tuning with original datasets.

We use the Adam optimizer with the cross-entropy loss as our loss function. We experiment with learning rates of $\{1e^{-3}, 1e^{-4}, 1e^{-5}, 1e^{-6}\}$ and choose the best one to report averaged performance results over 3 runs with different random seeds. We use 6 × GeForce RTX 2080 Ti GPU each with 11GB memory. The training process takes less than 1 hour to finish for all datasets.

**`BoW` Model** Similar with the BERT model, we use dev sets to select the best model during training. We employ early stopping with a patience of 5. This ensures a fair comparison between different settings.

We use the Adam optimizer with the cross-entropy loss as our loss function. We experiment with learning rates of $\{1e^{-3}, 1e^{-4}, 1e^{-5}\}$ and choose the best one to report averaged performance results over 3 runs with different random seeds. For Conditional Random Fields models (CRFs), we use `sklearn-crfsuite` library with default settings.[3] All models are trained using CPUs. The training process takes less than 15 minutes to finish for all datasets.

**`Dummy` Model** We use the dummy classifier in the `sklearn` library[4] with *stratified* strategy as our random model.

**Non-pretrained `BERT` Model** For training from scratch, we try two stop conditions. First, we employ early stopping with a patience of 5. Next, we also try another condition where we left the model to run for 500 epochs for every dataset except MRPC. For MRPC, we train it for 5000 epochs due to its small data size. We select the best performance out of these two options. This ensures the model to explore in the parameter space exhaustively and fair comparison between fine-tuned models and train-from-scratch models.

We use the BERT Adam optimizer with the cross-entropy loss as our loss function. We fix the initial learning rate at $1e^{-4}$, and choose the best one to report averaged performance results over 3 runs with different random seeds. We use 8 × GeForce RTX 2080 Ti GPU each with 11GB memory. The training process takes about 4 hours to finish for the largest dataset and 50 minutes for the smallest dataset. For the fixed epoch approach, the training process takes about 16 hours to finish for the largest dataset, and 5 hours for the smallest dataset.

## C   FREQUENCY MATCHING

To study the effect of word frequencies on the transferability of BERT, we control word frequencies when scrambling sentences. Figure 7 shows the differences in frequencies of matched pairs. Our results show that the difference in frequency for a frequency-matched pair is significantly smaller than a randomly matched pair.

---

[3] `https://sklearn-crfsuite.readthedocs.io/en/latest/`
[4] `https://scikit-learn.org/stable/modules/generated/sklearn.dummy.DummyClassifier.html`

To match word frequency during scrambling, we first preprocess sentences by lower-casing and separating by spaces and punctuation. We then use the original BERT WordPiece tokenizer to determine the sub-token length for each word, where the sub-token length is the number of word pieces a word contains. To randomly match words with similar frequencies, we first bucket words by their sub-token length. Then, we iterate through words within each bucket in the order of word frequencies. For each word, we use the round-robin method to find the closest neighbor with the closest frequency.

A perfect match is not always possible as not every word can be paired with another word with an identical word frequency. We include the distributions of the difference in frequency for every matched word pair in Appendix C to illustrate word frequencies are preserved.

## D  SCRAMBED SENTENCE

In Table 5 and Table 6, we provide one example sentence from each dataset destructed by our 4 scrambling methods. We also include the original English sentence (OR) at the top for each dataset.

| | Scrambling Method |
|---|---|
| the worst titles in recent cinematic history | Original Sentence |
| a engaging semi is everyone dull dark | Similar Frequency |
| kitsch theatrically tranquil andys loaf shorty lauper | Random |

(a) Scrambled Examples from the SST-3 dataset with different types of scrambling methods.

| | Scrambling Method |
|---|---|
| *premise:* a lady wearing a batman shirt is walking along the boardwalk . 
 *hypothesis:* a woman is swimming in a lake . | Original Sentence |
| *premise:* . car , . peach playing the outside hands is lay a 
 *hypothesis:* . with the baseball man . helmet a | Similar Frequency |
| *premise:* moist cleaver surf moist blades smurf hover bugger unto locals pinnies cotton 
 *hypothesis:* moist songs hover starves blacktop moist beam | Random |

(b) Scrambled Examples from the SNLI dataset with different types of scrambling methods.

| | Scrambling Method |
|---|---|
| *question:* what objects do musicians have to have in order to play woodwind instruments ? 
 *sentence:* despite their collective name, not all woodwind instruments are made entirely of wood . | Original Sentence |
| *question:* a pubs people bomb first and first , areas and october confessor witnesses of 
 *sentence:* video its rebels states in his world confessor witnesses ) under guam ? hall the | Similar Frequency |
| *question:* warranties mundine encountered froschwiller nir entering nir litatio pachomius entering mille says mc diaspora 
 *sentence:* mosfet bigua satisfactory merv gooding daewoo kennedy says mc iditarod scrofula depositing unprotected ubaidian oran | Random |

(c) Scrambled Examples from the QNLI dataset with different types of scrambling methods.

Table 5: Comparisons between the original English sentence and scrambled sentences.

| | Scrambling Method |
|---|---|
| *sentence1:* the court then stayed that injunction , pending an appeal by the canadian company . 

 *sentence2:* the injunction was immediately stayed pending an appeal to the federal circuit court of appeals in Washington . | Original Sentence |
| *sentence1:* . cents executive airways for simon to needs 1 economy from . custody no the 

 *sentence2:* . simon at loss airways needs 1 economy , . share sending cents in stores of dollar the | Similar Frequency |
| *sentence1:* najaf render analyzed threatening earners bethany hurlbert melville 517 riyadh birdie najaf hail weighs warden 

 *sentence2:* najaf bethany roared jackson threatening melville 517 riyadh eves najaf credentials manfred render mission noting deceptive things warden | Random |

(a) Scrambled Examples from the MRPC dataset with different types of scrambling methods.

| | Scrambling Method |
|---|---|
| relations with Russia , which is our main partner , have great importance " Kuchma said . | Original Sentence |
| overseas 0 NEW . are 4 city children Draw . after Wasim Mia . on turning 's | Similar Frequency |
| providing 585 soliciting Pushpakumara Grabowski dissidents Kuwait flick-on Sorghum Pushpakumara Goldstein Batty secure Pushpakumara 0#NKEL.RUO Gama 603 LUX | Random |

(b) Scrambled Examples from the EN-EWT dataset with different types of scrambling methods.

| | Scrambling Method |
|---|---|
| We walked in to pick our little man at 10 minutes to closing and heard laughter from kids and the staff . | Original Sentence |
| any murder is themselves good Iraq second my family Your hell a .? phenomenal n't death a . every the | Similar Frequency |
| northward Darfur Bert stink Minimum descriptive Ã³l gunning Turns discomfort TERRIBLE stink Washington passcode Ham's blurred human 15 passcode agree faction Goldman | Random |

(c) Scrambled Examples from the CoNLL-2003 dataset with different types of scrambling methods.

Table 6: Comparisons between the original English sentence and scrambled sentences.

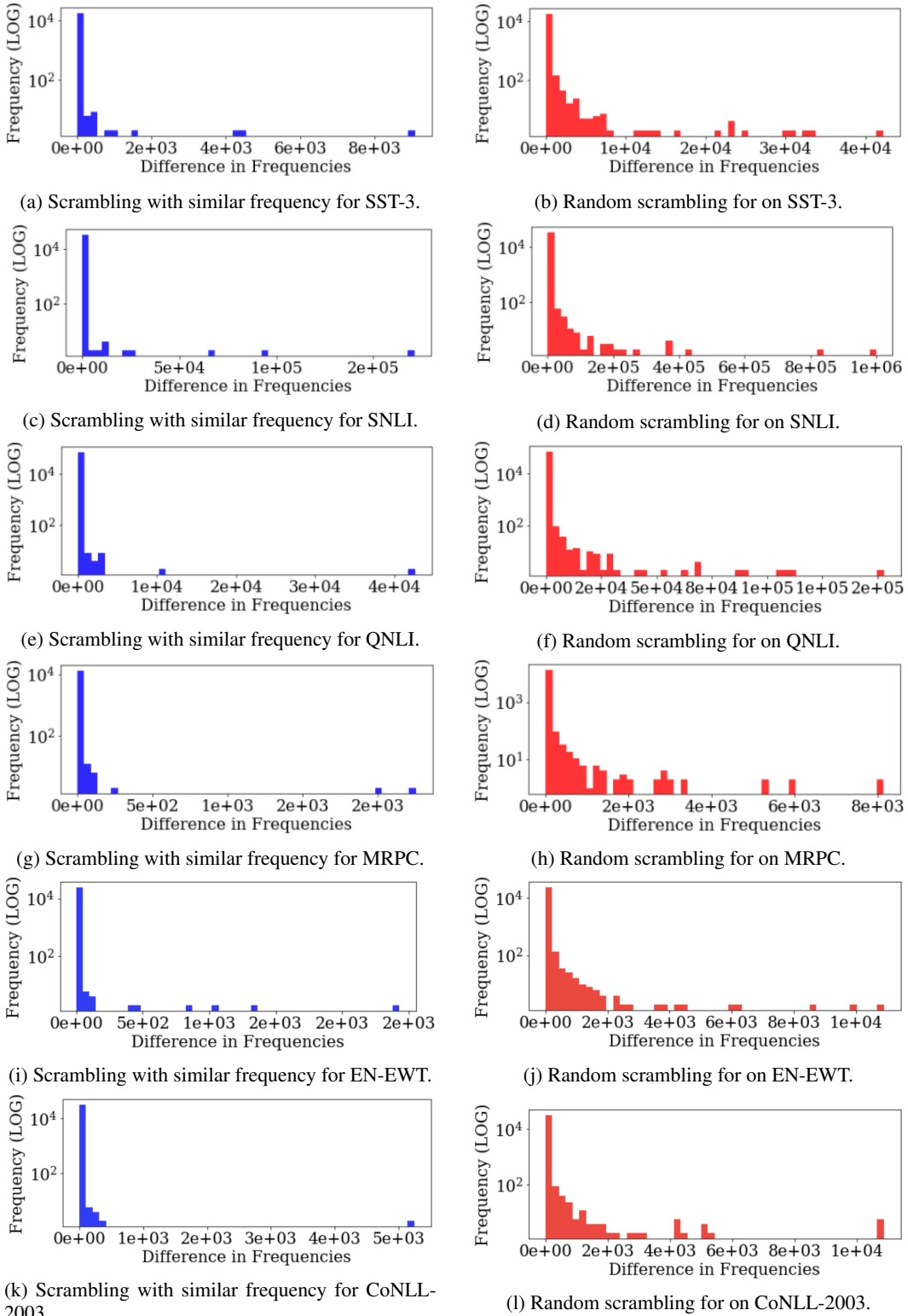

Figure 7: Distributions of difference in word frequency for each dataset.

