# OpenReview forum: "Identifying the Limits of Cross-Domain Knowledge Transfer for Pretrained Models"
_ICLR.cc/2022/Conference — ICLR 2022 Submitted_

### Official Review · Reviewer_6a8W · 2021-10-22

**Correctness:** 3
**Technical Novelty And Significance:** 2
**Empirical Novelty And Significance:** 2
**Recommendation:** 5
**Confidence:** 3

**Main Review:**

Strength:
1. the paper is mostly well-written and easy to follow.
2. the experiments are well-designed to support the claims

Weakness:
2. I'm not convinced by the results about BERT in S7.3. The confounder factor -- pre-training, is not considered. For instance, for the pre-trained BERT, the word frequencies information is already captured in the parameters (along with many other features that also hold for scrambled text). Such that when fine-tuned on scrambled data (word frequency preserved), it's not surprising that pre-trained BERT can clearly perform better. And the reason why LSTM is more robust is that the model size is very small such that it can relearn all weights. We can observe the same trend for 1-layer-BERT on classification tasks, the gaps are generally smaller than other model variants. Some claims in the paper can be further substantiated if the author can experiment with a BERT that is pre-trained on scrambled English.

3. I also have some doubts about S6.2, in which the author claim that the bad performance is because sequence labeling tasks are more likely to rely on word identities. However, as explored in Hewitt & Liang, those labeling supervisions (although in POS tagging) can be easily learned by fine-tuning. Can authors find a better explanation for the poor performance on labeling tasks?

1. The method proposed in this paper is widely used in many other studies, e.g., in adversarial attacking. But it is still somewhat novel since it is applied to new domains. Also, the observations made are marginally novel or significant, there are not many new results compared to prior observations. The title is about pre-trained models, but only BERT is discussed throughout the paper.

**Summary Of The Paper:**

This paper proposes an evaluation pipeline for pre-trained models by testing their transferability
without word identity information. Specifically, they take an English pre-trained BERT off-the-shelf
and fine-tune it with a corrupted English dataset. Those corrupted texts are designed to remove word identity information while preserving the word frequency information. They conduct experiments on 6 tasks.

**Summary Of The Review:**

This paper presents some interesting observations, but the methods or findings are only marginally significant or novel.

---

> ### Author Response · Authors · 2021-11-13
> **Helpful remarks for framing our results**
>
> Thank you for the comments and new ideas! They are helping us to think about our results in new ways.
>
> Regarding item 2 under “Weaknesses” – we’re not sure we follow the argument. The experiments in Section 7.1 show that word frequency statistics alone cannot account for the domain transfer effects we see, and the experiments in Section 7.5 show that BERT is not merely relearning the underlying scrambling function. Is there some other factor you have in mind that might be causing us to have overstated the finding in Section 7.3?
>
> > Some claims in the paper can be further substantiated if the author can experiment with a BERT that is pre-trained on scrambled English.
>
> Can you say more about what the paradigm would be? If BERT is simply trained from scratch in data that have been scrambling according to our scrambling function, then no meaningful change has been made, since these are bijective replacements that do not affect the co-occurrence statistics. Thus, we need not do this, since it will vary from regular BERT only in incidental ways. If the idea is to continue pretraining BERT on scrambled data and then fine-tune it in the ways we describe, then the paradigm is certainly different from what we do, but at the cost of reducing the value of the underlying BERT model, since its second phase of self-supervised pretraining is at odds with its first.  If there is a different variation of these experiments that you have in mind, please do let us know.
>
> > 3. I also have some doubts about S6.2, in which the author claim that the bad performance is because sequence labeling tasks are more likely to rely on word identities. However, as explored in Hewitt & Liang, those labeling supervisions (although in POS tagging) can be easily learned by fine-tuning. Can authors find a better explanation for the poor performance on labeling tasks?
>
> We’re certainly not saying that the models under discussion can’t do these  sequence labeling tasks. On the contrary, our results for “Standard” are very impressive. Our claim is only that there is less extreme cross-domain transfer for these tasks.
>
> > 4. The method proposed in this paper is widely used in many other studies, e.g., in adversarial attacking. But it is still somewhat novel since it is applied to new domains. Also, the observations made are marginally novel or significant, there are not many new results compared to prior observations.
>
> Beyond the prior work we cite in the text, we are not aware of any prior results for our particular problem. We’re pretty confident that the paradigm we’re using is novel. If there are papers we have overlooked, we are of course keen to know about them and incorporate them.
>
> > The title is about pre-trained models, but only BERT is discussed throughout the paper.
>
> Yes, agreed. We are open to expanding our results, perhaps via supplementary appendices. We have now run our main experiments on DeBERTa.. This comment provides those results:
>
> https://openreview.net/forum?id=luO6l9cP6b6&noteId=WwI4V7qFf16
>
> DeBERTa currently shows less transfer than BERT, but we plan to continue a hyperparameter search, and we expect that to bring DeBERTA more in line with BERT, since the architectures and protocols for these models are so similar.
>
> If there are other models that would be useful to assess in our current context, we would be happy to do that. Our fear is that additional exploration of Transformer-based model variants will not lead to substantially new results. Indeed, we woud be suspicious if they did seem to lead to radically new outcomes given how closely related all these models are.

---

> > ### Comment · Reviewer_6a8W · 2021-11-19
> > **RE**
> >
> > Thanks for the additional results and responses (i also read rebuttals for other reviewers), they help clear many misunderstandings and improve the paper. For point 4, upon a quick search, I already find one paper [1] that also performs token replacement based on word distribution. Token replacement is a very common practice in adversarial attacking and data augmentation, maybe the author can further check some surveys to avoid possible conflicts.
> >
> > It would also be great if the author can add a few more paragraphs to talk about how findings in this paper can benefit future research.
> >
> > Overall, I think the paper has been improved after rebuttal. And I would be happy to accept it as a full paper in a good workshop, or a short paper at good conferences, but not as a full paper at ICLR yet since the contribution is quite niche and focused.
> >
> >
> > [1] Generating Adversarial Text Samples. ECIR 2016

---

> > > ### Author Response · Authors · 2021-11-19
> > > **Very different methods and goals from 'Generating Adversarial Text Samples'**
> > >
> > > > For point 4, upon a quick search, I already find one paper [1] that also performs token replacement based on word distribution.
> > >
> > > Thanks for the pointer. That nice paper, 'Generating Adversarial Text Samples', performs word substitutions for subsets of the tokens, the substitutions are deliberately crafted to maintain coherence, and the goal is to find weaknesses in trained models.
> > >
> > > By contrast, we perform total vocabulary-level substitutions that result in sentences that no one could understand without first decoding them using the scrambling function's look-up table, and our goal us to show that fine-tuning on such data shows a surprising amount of transfer.

---

> > > ### Author Response · Authors · 2021-11-19
> > > **The importance of understanding the pretraining + fine-tuning paradigm**
> > >
> > > > the contribution is quite niche and focused.
> > >
> > > Our view is that pretraining + fine-tuning is arguably the most mainstream thing one can do right now, throughout many parts of AI, and so anything we can do to better understand the nature and limits of that method is fundamentally important.

---

### Official Review · Reviewer_dnE4 · 2021-11-01

**Correctness:** 4
**Technical Novelty And Significance:** 2
**Empirical Novelty And Significance:** 3
**Recommendation:** 6
**Confidence:** 4

**Main Review:**

Overall, I think the paper is well-written and debates interesting points in transferability.
However, I want to see some measures of intrinsic evaluation related to the scrambled datasets
as well (e.g., perplexity). My concern is that these scrambling techniques might push your
dataset outside of real word adversarial attacks. I would suggest comparing them all together
to support validity of the scrambling methods. Furthermore, could you every tried replacing
word using ontology or a collection of semantically related words? Please add similar
experiment to show that scrambling with synonyms and antonym have different effects (or
positive vs negative replacements).

**Summary Of The Paper:**

This paper elaborates on transfer learning and domain adaptation in language models. The
authors argue that there are limits on how much information could be transferred when a
model is scrambled or somehow randomized. They used different strategies to randomize
training data including frequency matching and completely random replacement. They suggest
that only Bert shows high rates of transfer into their scrambled domains in classification tasks.
Their experimental results exhibit importance of pretraining on sequence labeling tasks where
randomness occurs. For example, they showed that identity of words are important and when
they are swapped with completely different frequencies (random) the performance drops
significantly. They also supported the view of Sinha et al. (2021) and Ethayarajh (2019) that BERT
model may preserves frequency better and this is the reason behind its superiority.

**Summary Of The Review:**

I believe regardless  of how good some of transfer learning models perform, we need to elaborate on situations those models collapse. Therefore I think having similar papers truly exhibit weakness and strengths of these models is necessary.

---

> ### Author Response · Authors · 2021-11-13
> **Valuable new ideas for metrics and experiments**
>
> Thank you very much for these remarks. They suggest some important new directions for the work that we are already pursuing.
>
> > However, I want to see some measures of intrinsic evaluation related to the scrambled datasets as well (e.g., perplexity).
>
> Excellent idea. We will add a full report on perplexity to the paper. Our current plan is to report perplexity on the dev sets for our GLUE tasks. These findings might need to be in an appendix due to space limitations, but we can point to it in the main text. Here is an initial set of results for BERT, for all of GLUE:
>
> |      |   Standard |   Scramble: Random |   Scramble: Similar Freq |
> |:-----|-----------:|-------------------:|-------------------------:|
> | cola |      28.02 |              35.88 |                  2161.03 |
> | mnli |      36.51 |              60.76 |                  1618.86 |
> | mrpc |      22.48 |              59.94 |                  1596.88 |
> | qnli |      48.24 |              92.21 |                  2245.72 |
> | qqp  |      36.69 |              75.56 |                  2613.05 |
> | rte  |      13.93 |              47.77 |                  1210.45 |
> | sst2 |      22.97 |              57.96 |                  1277.03 |
> | stsb |      30.27 |              52.27 |                  2012.56 |
> | wnli |      16.7  |              32.48 |                  1310.35 |
>
> It’s illuminating that Similar Frequency has so much higher perplexity than Random. In matching frequency, we really challenge these models!
>
> >  My concern is that these scrambling techniques might push your dataset outside of real word adversarial attacks.
>
> We are interested in this comment but not sure what it is getting at. We don’t think of our experiments are related to adversarial testing or adversarial attacks. Rather, they serve to highlight the robust nature of domain transfer in pretraining + fine-tunign paradigms. If there is an experimental angle related to adversarial attacks, it might be illuminating, though. Any additional thoughts here would be appreciated!
>
> > Please add similar experiment to show that scrambling with synonyms and antonym have different effects (or positive vs negative replacements).
>
> This seems like an excellent idea. We are currently thinking through the details of how this would be done, in terms of which lexical resources to rely on. We will post an update when we have a proposal for an experimental protocol and some pilot results for SST.

---

### Official Review · Reviewer_4jKu · 2021-11-01

**Correctness:** 3
**Technical Novelty And Significance:** 1
**Empirical Novelty And Significance:** 1
**Recommendation:** 5
**Confidence:** 5

**Main Review:**

The paper is well-written and does a reasonable job evaluating the robustness of architectures under an artificial form of vocabulary shift. However, the authors fail to convince me why studying this distributional shift would bring new and relevant insights to the table. I think for two reasons:

(a) The shift is not interesting in the sense that it is related to cross-domain or cross-lingual or cross-time shifts, or any other shifts that we observe in the wild.
(b) The shift does not seem to add value over other synthetic shifts that have already been studied, e.g., word-level shuffling, character-level shuffling, etc.

You could have studied many other scrambling functions, so why this one? That said, I liked §7.3, which nicely baselines the scrambled model performance. §7.5 is also nice, except that it’s hard to see why any network would learn to reassociate words in the context of learning a particular task. (Learning to reassociate words is not necessary to solve most tasks. Maybe in machine translation?)

Writing: The paper is overall well-written, but I found the use of the word ‘scrambling’ for a left-to-right in-order replacement of words, misleading. In related work, scrambling refers to scrambling of word order. I would also have liked to see a stronger motivation for focusing on frequency.

Other weaknesses:

(i) BERT is old now. We have already seen a lot of studies of BERT (in the so-called ‘Bertology’-literature), but isn’t it time to move on? I’m a little worried about community-wide overfitting of our intuitions about language models. There’s the popular alternatives (RoBERTa, GPT-2, t5, etc.), but also a lot of faster, fairer, more interpretable alternatives.

(ii) The two sequential labelling tasks are very similar. POS is a give-away of NER. How about finegrained sentiment, coreference resolution, semantic tagging, grammatical error detection, etc.?

(iii) As shown in related work, the GLUE tasks are relatively easy (e.g., solvable without word order information). Maybe also consider harder sequence classification problems?

(iv) Using only a small subset of GLUE tasks looks like cherry-picking.

(v) It is unclear how sensitive the results for each architecture are to hyper-parameter changes.

**Summary Of The Paper:**

The paper studies the generalization of several architectures (BOW, GloVe, LSTM, BERT) under scrambling of vocabularies - with and without frequency constraints.

**Summary Of The Review:**

The paper is well-written, but poorly motivated. I simply do not see what we learn from the paper (in the intersection of 'previously unknown' and 'relevant'). In addition, while the experiments are well-designed, the datasets seem cherry-picked, and the models out-dated.

---

> ### Author Response · Authors · 2021-11-13
> **Full GLUE results**
>
> > (iv) Using only a small subset of GLUE tasks looks like cherry-picking.
>
> That’s fair! We don’t have space in our main text for the full set of GLUE results, but we will add an appendix giving results comparable to Table 2, or else expand Table 2 and then do our analytic experiments on our current subset of tasks, so that we can fit everything in.
> Here are the full GLUE results:
>
> |         |   BERT – Standard |   BERT – Scramble: Random |   BERT – Scramble: Similar Freq | LSTM+GloVe – Standard   | BoW – Standard   |
> |:--------|------------------:|--------------------------:|--------------------------------:|------------------------:|-----------------:|
> | cola    |              0.56 |                      0.09 |                            0.12 | 0.16                    | 0.0              |
> | mnli    |              0.85 |                      0.7  |                            0.71 | 0.7                     | 0.56             |
> | mnli-mm |              0.85 |                      0.7  |                            0.72 | -                       | -                |
> | mrpc    |              0.85 |                      0.77 |                            0.76 | 0.69                    | 0.73             |
> | qnli    |              0.92 |                      0.79 |                            0.79 | 0.76                    | 0.72             |
> | qqp     |              0.91 |                      0.87 |                            0.88 | 0.82                    | 0.79             |
> | rte     |              0.68 |                      0.55 |                            0.56 | 0.53                    | 0.54             |
> | sst2    |              0.93 |                      0.83 |                            0.81 | 0.86                    | 0.8              |
> | stsb    |              0.89 |                      0.78 |                            0.78 | 0.66                    | 0.61             |
> | wnli    |              0.13 |                      0.25 |                            0.14 | 0.65                    | 0.65             |
>
> The result for WNLI are a bit surprising, but that is a small, high variance NLI task, so we are not sure that there is much to investigate there.

---

> > ### Comment · Reviewer_4jKu · 2021-11-18
> > **Re: Full GLUE results**
> >
> > Thanks a ton, maybe consider a discussion of the COLA results in the paper/appendix?

---

> > > ### Author Response · Authors · 2021-11-18
> > > **Yes, perhaps in the main text!**
> > >
> > > We are looking for ways to fit the full GLUE table into the main text and discuss the variation we see at a high-level, relating it to what we know about the tasks from prior work. Due to space constraints, our subsequent experimental sections will report on only a subset of these tasks (plus the others that aren't in GLUE but make other points for us). However, we can put all those results in an appendix.

---

> ### Author Response · Authors · 2021-11-13
> **Choice of scrambling functions**
>
> > (b) The shift does not seem to add value over other synthetic shifts that have already been studied, e.g., word-level shuffling, character-level shuffling, etc.
>
> We should add discussion of this in the paper -- it’s a really nice framework for thinking about the motivations for our experiments. In brief:
>
> 1. Word-level shuffling: this tests how sensitive the model is to word order and possibly also syntactic (phrase-level) associations.
> 2. Character-level shuffling: Depending on the model’s tokenization scheme, this might test the same thing that word-level shuffling does, or it might test how much the model is able to recover word-level units from a character-level tokenization.
> 3. Our paradigm: This assesses the extent of transfer even where no information about word identity is available.
>
> We think 3 is very different from 1 and 2 in terms of the conclusions that can be supported.
>
> > You could have studied many other scrambling functions, so why this one?
>
> This is the most direct manipulation we could think of that removes word-identity information. That said, we are open to exploring other functions.

---

> ### Author Response · Authors · 2021-11-13
> **Going beyond BERT evaluations**
>
> > (i) BERT is old now. We have already seen a lot of studies of BERT (in the so-called ‘Bertology’-literature), but isn’t it time to move on? I’m a little worried about community-wide overfitting of our intuitions about language models. There’s the popular alternatives (RoBERTa, GPT-2, t5, etc.), but also a lot of faster, fairer, more interpretable alternatives.
>
> This is fair. We picked BERT as the most commonly used model for this kind of pretraining + fine-tuning paradigm. To the extent that BERT is used pervasively for this, it is important for us to understand the nature of the process. The other models you mention are also probably successful in pretraining + fine-tuning, and so we can ask the same questions about them. Our methods are either purely behavioral or depend only on having multiple layers in the model, so they are widely applicable to different settings where people do pretraining + fine-tuning.
>
> We have now run our main experiments on DeBERTa and will report those in an appendix. This comment provides those results:
>
> https://openreview.net/forum?id=luO6l9cP6b6&noteId=WwI4V7qFf16
>
> DeBERTa currently shows less transfer than BERT, but we plan to continue a hyperparameter search, and we expect that to bring DeBERTA more in line with BERT, since the architectures and protocols for these models are so similar.
>
> If there are other models that would be useful to assess in our current context, we would be happy to do that. Our fear is that additional exploration of Transformer-based model variants will not lead to substantially new results. Indeed, we woud be suspicious if they did seem to lead to radically new outcomes given how closely related all these models are.

---

> > ### Comment · Reviewer_4jKu · 2021-11-18
> > **Re: Going beyond BERT evaluations**
> >
> > Thanks for your additional results. On a general note, if you inspect transfer to be constant across models, why is it then interesting in the first place? This is motivated as exploring the limits of domain transfer, but aren't we interested in what enables/inhibits transfer? It would be really interesting if you could correlate differences in transfer on your artificial task (between runs, models, or architectures) with downstream properties of general interest, i.e., generalization, interpretability, fairness, or privacy. Not sure what to recommend, but my knee jerk would be to consider learning objective, positional encodings, and maybe model size? In my view, you need to show edge case behavior on vocabulary shift tells us something interesting about model behavior in the wild, and that it is a reason to choose some models over others.

---

> > > ### Author Response · Authors · 2021-11-18
> > > **Transfer is interesting for this incredibly important class of related models**
> > >
> > > > On a general note, if you inspect transfer to be constant across models, why is it then interesting in the first place?
> > >
> > > Fine-tuning the models in question is one of the most common things in the literature right now, driving huge numbers of results and large research efforts. Many of the models are very similar in structure, which generally leads them to produce similar fine-tuning results, but this doesn't reduce the importance of understanding the extent of the abilities these models have to achieve within-domain and cross-domain transfer.
> > >
> > > Our paper shows that LSTM+GloVe does not show cross-domain transfer in our sense, and our results where we freeze different layers of BERT show pretty clearly that smaller networks show less transfer. We are totally open to testing other models, but we're not sure what hypothesis this would be testing – presumably this would relate to something about the model's particular structure and known capacity for fine-tuning. This would likely open up a new strand of inquiry on top of our results.

---

> > > ### Author Response · Authors · 2021-11-18
> > > **Important new and complementary directions**
> > >
> > > >  It would be really interesting if you could correlate differences in transfer on your artificial task (between runs, models, or architectures) with downstream properties of general interest, i.e., generalization, interpretability, fairness, or privacy. Not sure what to recommend, but my knee jerk would be to consider learning objective, positional encodings, and maybe model size? In my view, you need to show edge case behavior on vocabulary shift tells us something interesting about model behavior in the wild, and that it is a reason to choose some models over others.
> > >
> > > These are certainly worthwhile questions! Our experiments in section 7.4 (and 7.5, in part) help address the model size issue. For the other questions, it seems like they could fill a separate paper, and that paper's findings would leave open the questions we address concerning (1) whether the transfer we see reduces to word frequency statistics (it doesn't), (2) whether pretraining is crucial vs. simply fine-tuning (it is), and (3) whether the transfer is simply the result of "unscrambling" the scrambling function (it isn't).

---

> ### Author Response · Authors · 2021-11-13
> **Expanding on the motivations and findings**
>
> We really appreciate this incisive feedback on our motivations and findings. It’s giving us a chance to reflect on how we might provide broader context for the work and, in this way, help to convey the relevance of our findings.
>
> > (a) The shift is not interesting in the sense that it is related to cross-domain or cross-lingual or cross-time shifts, or any other shifts that we observe in the wild.
>
> We would like to embrace the fact that it’s not a shift we would see in the wild, but rather one that we can study with real analytical precision, since our experimental paradigm gives us so much control. For example, ohe experiments completely factor out word identity information and the associated simple distributional statistics of such items, which one might expect to be the sole driver of transfer. Experiments on naturalistic corpora would not support such direct findings.
>
> > it’s hard to see why any network would learn to reassociate words in the context of learning a particular task.
>
> Interesting! This is fair. In talking about our work with colleagues, they have often raised the question of whether the model simply re-associates words in this way, so we pursued the hypothesis.
>
> > The paper is overall well-written, but I found the use of the word ‘scrambling’ for a left-to-right in-order replacement of words, misleading.
>
> Thanks for this! Yes, we share your intuition about “scrambling” as a term/concept. We can try to find a new label for the manipulation. We need something that conveys that this is “vocabulary-level shuffling” or the like.
>
>
> > (ii) The two sequential labelling tasks are very similar. POS is a give-away of NER. How about finegrained sentiment, coreference resolution, semantic tagging, grammatical error detection, etc.?
>
> Sure! This would definitely add to the paper, and we might be able to fit it. Fine-grained sentiment doesn’t have very many high-quality datasets in our view. Coref has different patterns than POS tagging in probing experiments, so that could be good. Grammatical error detection would require some deep thinking about methods in the context of our scrambling technique.
>
> > (iii) As shown in related work, the GLUE tasks are relatively easy (e.g., solvable without word order information). Maybe also consider harder sequence classification problems?
>
> We feel there is not space in the paper for more experiments. Harder classification tasks could be illuminating, but the important thing is that our current set of tasks provide plenty of space (in terms of relative performance) to observe differences between regular fine-tuning, our scrambled fine-tuning, and our baseline LSTMs.
>
> > (v) It is unclear how sensitive the results for each architecture are to hyper-parameter changes.
>
> We do not observe a lot of sensitivity, but we will try to quantify this more precisely and add some details to our methods appendix.

---

### Official Review · Reviewer_vdgZ · 2021-11-05

**Correctness:** 2
**Technical Novelty And Significance:** 3
**Empirical Novelty And Significance:** 3
**Recommendation:** 3
**Confidence:** 4

**Main Review:**


The overall presentation is regular. I think authors fail to properly state all the possible explanations to support the existence of transfer learning with scrambled or random input,
If transfer learning were the explanation of why scrambled input keeps showing transfer learning capabilities an experiment training with the original data while tested on scrambled or vice versa is needed. Why didn't the authors try that configuration?
The analysis section can be largely improved. Three out of four experiments show inconclusive results. If results cannot be interpreted, authors should find other experiments to support (or refute) their claims.
Despite the interesting results and large number of experiments, the inconclusive results make unclear the scientific contribution of this research piece.

Some details:

_Paper's title is somehow misleading "Cross-Domain Knowledge Transfer for Pretrained Models". Authors use just BERT as pretrained model and a LSTM with GloVe embeddings as part of the experimentation.


_ The introduction and the Conclusions are inconsistent:
"we evaluate LSTMs using GloVe embeddings, BERT, and baseline models" vs "we take an English pretrained BERT off-the-shelf and fine-tune it with a scrambled English dataset"


_table 4 is shown on page 6, but referenced on page 8. Please bring the table closer to its reference.

_ "Our leading hypothesis here is that the LSTMs may actually relearn all weights without taking advantage of pretraining".


**Summary Of The Paper:**

This work aims to identify the source of transfer learning on neural models.  To this end they set up a series of experiments where models are trained and tested with english data, and, with scrambled (or randomly replaced) data, keeping token-wise sentence length.
Scrambled sentences also use replacement tokens keeping token frequency with the original token.
Models are tested on standard benchmarks on text classification and sequence modelling, showing a drop in performance when the input is scrambled, and a similar drop with randomized input.
Further analysis are meant to test of the scrambling maintains sentence semantics (it doesn't'), if BERT is just retraining (there is some transfer but inconclusive), keeping BERT frozen (it depends on the layer) and finally identifying word identity reassociation (it seems there is not reassociation).


**Summary Of The Review:**

Finding are unclear
The paper is difficult to follow
Writing can be improved
Experiments are inconclusive.

---

> ### Author Response · Authors · 2021-11-13
> **New zero-shot transfer experimental results**
>
> On the question of why we didn’t do
>
> > an experiment training with the original data while tested on scrambled or vice versa is needed”
>
> This is an interesting paradigm. It provides a test of models’ capacity to do zero-shot generalization across the scrambled/unscrambled boundary. By contrast, our paper is devoted to helping the community better understand where and how task-specific fine-tuning of these models is effective.
>
> That said, we did run these zero-shot experiments. As one might expect, they establish a lower-bound on the amount of transfer we could possibly see, since there is no fine-tuning. Here are those results for the entire GLUE benchmark:
>
> |         |   BERT – Standard |   BERT – Scramble: Random |   BERT – Scramble: Similar Freq |   BERT (zero-shot) – Scramble: Random |   BERT (zero-shot) – Scramble: Similar Freq | LSTM+GloVe – Standard   | BoW – Standard   |
> |:--------|------------------:|--------------------------:|--------------------------------:|--------------------------------------:|--------------------------------------------:|------------------------:|-----------------:|
> | cola    |              0.56 |                      0.09 |                            0.12 |                                 -0.05 |                                        0    | 0.16                    | 0.0              |
> | mnli    |              0.85 |                      0.7  |                            0.71 |                                  0.37 |                                        0.41 | 0.7                     | 0.56             |
> | mnli-mm |              0.85 |                      0.7  |                            0.72 |                                  0.38 |                                        0.39 | -                       | -                |
> | mrpc    |              0.85 |                      0.77 |                            0.76 |                                  0.65 |                                        0.72 | 0.69                    | 0.73             |
> | qnli    |              0.92 |                      0.79 |                            0.79 |                                  0.52 |                                        0.65 | 0.76                    | 0.72             |
> | qqp     |              0.91 |                      0.87 |                            0.88 |                                  0.63 |                                        0.67 | 0.82                    | 0.79             |
> | rte     |              0.68 |                      0.55 |                            0.56 |                                  0.55 |                                        0.6  | 0.53                    | 0.54             |
> | sst2    |              0.93 |                      0.83 |                            0.81 |                                  0.49 |                                        0.52 | 0.86                    | 0.8              |
> | stsb    |              0.89 |                      0.78 |                            0.78 |                                  0.6  |                                        0.73 | 0.66                    | 0.61             |
> | wnli    |              0.13 |                      0.25 |                            0.14 |                                  0.44 |                                        0.48 | 0.65                    | 0.65             |
>
> We can plan to put these in an appendix. The zero-shot cases are very, very bad, as one would expect given the extreme domain shift. This makes our positive results where there is fine-tuning all the more remarkable.

---

> ### Author Response · Authors · 2021-11-13
> **Additional points**
>
> > _ The introduction and the Conclusions are inconsistent: "we evaluate LSTMs using GloVe embeddings, BERT, and baseline models" vs "we take an English pretrained BERT off-the-shelf and fine-tune it with a scrambled English dataset"
>
> The baseline moldes are a bag of word model and a random model. They are included to provide context for our results. We can mention them again in the conclusion, but we need to ensure that the focus is on the comparison between BERT and the LSTM, with the others playing only general supporting roles.
>
> > _table 4 is shown on page 6, but referenced on page 8. Please bring the table closer to its reference.
>
> We will work to make the layout better in this regard.
>
> > _ "Our leading hypothesis here is that the LSTMs may actually relearn all weights without taking advantage of pretraining".
> Yes, this sentence appears in Section 7.3, and it is robustly supported by those experiments. It helps to show how important pretraining is to the kinds of transfer we see with BERT.
>
> > Writing can be improved
>
> We worked hard on the paper to make it clear, but of course we may have have fallen short. We will review the entire text with this in mind. If there are specific passages that you found to be poorly written, we’d appreciate hearing about those so we can make improvements.

---

> > ### Comment · Reviewer_vdgZ · 2021-11-18
> > **The introduction and the Conclusions are inconsistent**
> >
> > The paper abstract and the introduction should invite the reader to continue reading the paper, proposing a set of claims that will be proven via mathematical formulations or empirical results in the rest of the work. The conclusions should acknowledge that your initial claims were demonstrated and they should also list a set of findings the authors (might have) not expected originally.
> >
> > I invite the authors to revisit their introduction and conclusions
> > "We offer a partial answer via a systematic exploration of how much transfer occurs when models are denied any information about word identity via random scrambling. In four classification tasks and two sequence labeling tasks, we evaluate LSTMs using GloVe embeddings, BERT, and baseline models."
> > While in the conclusions you mention that:
> > "In this paper, we propose an evaluation pipeline for pretrained models by testing their transferability without word identity information. Specifically, we take an English pretrained BERT off-the-shelf and fine-tune it with a scrambled English dataset"
> >
> > I'm not saying the two sentences are not related, but if the objective of the work was to propose an evaluation pipeline, the authors should have been stated that in the intro (and maybe in the title)

---

> > > ### Author Response · Authors · 2021-11-18
> > > **Useful suggestion!**
> > >
> > > Sure, we are happy to revise these passages to make them clearer! Thanks!

---

> > ### Comment · Reviewer_vdgZ · 2021-11-18
> > **Our leading hypothesis here is that the LSTMs may actually relearn all weights without taking advantage of pretraining**
> >
> > I think my comment was incomplete in the review.
> > The sentence in section 7.3 refers to the table 4 which shows a similar performance of the LSTM with or without GloVe embeddings. If the replacements are done with any similar in frequency word, this results is expected. In this particular configuration I'm interested what would happen to scrambled the words keeping as similar as possible the similarity among the words.

---

> > > ### Author Response · Authors · 2021-11-18
> > > **Scrambling strategies and what they can tell us**
> > >
> > > Scrambling without frequency similarity is our "Random scrambling" strategy. We show that it is not shaped by frequency, so frequency is not a factor in the results seen here.
> > >
> > > If we scramble while maintaining similarity for other words (or for the swapped ones), then there is less need for domain transfer and so the results will certainly be better than for our two scrambling strategies. This follows from the role that representational similarity plays pervasively in these networks.

---

> ### Author Response · Authors · 2021-11-13
> **Evaluating additional models**
>
> Regarding the additional points raised:
>
> > _Paper's title is somehow misleading "Cross-Domain Knowledge Transfer for Pretrained Models". Authors use just BERT as pretrained model and a LSTM with GloVe embeddings as part of the experimentation.
>
> Our experiments focus on BERT, but our paper provides a set of experimental paradigms that apply much more widely. However, we are open to making the title more focused, to clarify that the experiments are on BERT.
>
> In addition, we have now run our main experiment on all of GLUE using DeBERTa. Here are those results:
>
> |         |   DeBERTa – Standard |   DeBERTa – Scramble: Random |   DeBERTa – Scramble: Similar Freq | LSTM+GloVe – Standard   | BoW – Standard   |
> |:--------|---------------------:|-----------------------------:|-----------------------------------:|------------------------:|-----------------:|
> | cola    |                 0.61 |                         0.07 |                               0.11 | 0.16                    | 0.0              |
> | mnli    |                 0.87 |                         0.66 |                               0.66 | 0.7                     | 0.56             |
> | mnli-mm |                 0.87 |                         0.68 |                               0.67 | -                       | -                |
> | mrpc    |                 0.88 |                         0.72 |                               0.76 | 0.69                    | 0.73             |
> | qnli    |                 0.92 |                         0.78 |                               0.79 | 0.76                    | 0.72             |
> | qqp     |                 0.9  |                         0.84 |                               0.85 | 0.82                    | 0.79             |
> | rte     |                 0.66 |                         0.54 |                               0.51 | 0.53                    | 0.54             |
> | sst2    |                 0.94 |                         0.72 |                               0.66 | 0.86                    | 0.8              |
> | stsb    |                 0.88 |                         0.68 |                               0.7  | 0.66                    | 0.61             |
> | wnli    |                 0.56 |                         0.56 |                               0.56 | 0.65                    | 0.65             |
>
> They show slightly less transfer than BERT overall, but we plan a wider sweep of hyperparameter tuning, and we expect that to yield results that are like those of BERT, since these two architectures are so similar.
>
> If there are other models that would be useful to assess in our current context, we would be happy to do that. Our fear is that additional exploration of Transformer-based model variants will not lead to substantially new results.

---

> > ### Comment · Reviewer_vdgZ · 2021-11-18
> > **Pretrained language model**
> >
> > DeBerta is certainly a different LM from BERT, but I would rather prefer to see results on LM which are more similar to BERT such as different BERT sizes (large vs base), RoBERTa or ALBERT, or a distilled version of it. Given that DeBerta changes the way the input sentences are represented "Unlike BERT where each word in the input layer is represented using a vector which is the sum of its word (content) embedding and position embedding, each word in DeBERTa is represented using two vectors that encode its content and position", I'm not sure if the results with DeBERTa are useful to support your claims.
> > And doesn't erase the original complaint, experimenting with a single pretrained language model is not enough to title the paper as a analysis on "Pretrained Models"

---

> > > ### Author Response · Authors · 2021-11-18
> > > **Diverse models are more informative**
> > >
> > > Our intuition is that we will learn more about the space by evaluating models that are *more* different from each other, not less. A lot of evaluations on closely related models seems likely to lead to a lot of very similar outcomes, whereas evaluating a wide range of model is more likely to teach new and different lessons. We are certainly open to doing such evaluations, but it is unlikely to fit into the main text of our already very full paper, so these results will probably be appendix results for now.

---

> ### Author Response · Authors · 2021-11-13
> **Experimental results are far from inconclusive**
>
> We appreciate the critical feedback on our paper. It has provided us with a useful opportunity reflect on how we are characterizing our goals and findings.
>
> We’d like to first address the statement that “Three out of four experiments show inconclusive results.” This is surprising to us as a summary of the experimental findings, none of which seem inconclusive to us. To summarize:
>
> 1. The “Frequency Effects” experiments in Section 7.1 rule out the simple hypothesis that the transfer effects we see reduce to simple effects of word frequency distributions. We take this experiment to show that that does not provide a complete explanation.
> 2. The “Does Scrambling Preserve Meaning?” experiments (Section 7.2) rule out the hypothesis that our scrambling procedure is preserving meaning in ways that would not really require any transfer of knowledge.
> 3. The “Transfer or simple pretraining?” experiments (Section 7.3) rule out the idea that this is merely retraining. There is real transfer.
> 4. The “Frozen BERT parameters” experiments (Section 7.4) show that pretraining is a component in transfer, but that it interacts with the fine-tuning paradigm.
> 5. The “Probing for Word Identity Reassociations” experiments (Section 7.5) show that models are not simply learning to “unscramble” our scrambling procedure. We find no evidence of this.
>
> Taken together, these experiments show that the transfer effects are not merely artifacts of the data or the model’s capacity to track simple first-order statistics of the data. They run much deeper and are directly connected to the very large representational power that these models have, and they show that pretraining is a non-trivial part of the equation – i.e., there really is non-superficial transfer.

---

### Author Response · Authors · 2021-11-18
**New Embedding Reinitialization experiments**

The reviewers’ comments led us to see that our scrambling function can be seen as a way to reinitialize the word embeddings by swapping embeddings between words randomly. Thus, to complement our identity swapping, we plan to add a new set of experiments where we directly reinitialize the weights of the embedding layer before fine-tuning. We have found strikingly different fine-tuning results with this approach. For example, here are the BERT results for all of GLUE (DeBERTa looks similar):

|         |   BERT – Standard |   BERT – Scramble: Random |   BERT – Scramble: Similar Freq |   BERT – emb_reinit |
|:--------|------------------:|--------------------------:|--------------------------------:|--------------------:|
| cola    |              0.56 |                      0.09 |                            0.12 |                0.08 |
| mnli    |              0.85 |                      0.70  |                            0.71 |                0.32 |
| mnli-mm |              0.85 |                      0.70  |                            0.72 |                0.32 |
| mrpc    |              0.85 |                      0.77 |                            0.76 |                0.63 |
| qnli    |              0.92 |                      0.79 |                            0.79 |                0.51 |
| qqp     |              0.91 |                      0.87 |                            0.88 |                0.63 |
| rte     |              0.68 |                      0.55 |                            0.56 |                0.53 |
| sst2    |              0.93 |                      0.83 |                            0.81 |                0.8  |
| stsb    |              0.89 |                      0.78 |                            0.78 |                0.38 |
| wnli    |              0.13 |                      0.25 |                            0.14 |                0.56 |

Given this contrast between word identity swapping (the `Scramble` conditions) and embedding reinitialization (`embed_reinit`), we can see that, although both diminish the model’s knowledge about word identities, the `Scramble` conditions still maintain 80%-90% of standard fine-tuning performance for most of tasks, whereas `embed_reinit` fails catastrophically. One key difference here is that `Scramble` maintains embeddings in the BERT space while `embed_reinit` scatters the embeddings randomly in high-dimensional space.

Corroborating our results in Section 7.5 (Probing for Word Identity Reassociation), these new results suggest that although `Scramble` functions destroy word identities, they still give the model good initializations to optimize for downstream tasks. This fact also informs our thinking about pre-trained language models like BERT are encoding semantics, or are providing good initializations for optimization in terms of sentence classification tasks.

---

### Author Response · Authors · 2021-11-18
**Broader impacts for claims about natural language understanding**

Thinking about our reviews has also led us to identify a new area of significance for our findings that we plan to highlight in our revision:

If pre-trained models can transfer knowledge without word identities with high performance, should we start to rethink how informative this high performance is when it comes to claims about what these models understand about language?

Benchmarks that are created for evaluating natural language understanding need to consider whether models are truly understanding semantics. Our results suggest that a large percentage of this success might trace to factors that have nothing to do with communication or understanding. After all, our scrambled data do not have the semantics of English, or indeed of any language, even at the most fundamental level (the lexicon or tokens in context).

Our paper concentrates on the positive lesson our results teach about cross-domain transfer, but this more pessimistic conceptual finding is important as well.

---

### Decision · Program_Chairs · 2022-01-20

**Decision:**

Reject

**Comment:**

This work performs an analysis of the generalization ability of pre-trained models under the condition of vocabulary scrambling. The paper is well written and easy to understand. However, a full story and investigation into the cause of the observed transfer under word scrambling is lacking. For example, do more powerful models transfer less because pre-training is more effective?

While the effect described in the paper is interesting, it lacks a solid connection to important areas such as adversarial attacks, cross-lingual domain shifts and doesn't seem to have any effect on the application of fine-tuning to pre-trained models. The experimental section could also be improved with comparisons to other more recent models such as RoBERTa and GPT-2. Even though these more recent models are still Transformer-based models it can help answer the question if more powerful models transfer less under word scrambling as raised by reviewer 6a8W. The results on LSTMs seem to imply this case. We thank the authors for including additional results on DeBERTa but this was insufficient to change the authors' opinion of the value of the paper.